# Methyladenosine Modification in RNAs: From Regulatory Roles to Therapeutic Implications in Cancer

**DOI:** 10.3390/cancers14133195

**Published:** 2022-06-29

**Authors:** Xiaolin Qu, Yongqiu Zhang, Xianzheng Sang, Ding Ren, Hong Zhao, Stephen T. C. Wong

**Affiliations:** 1Department of Neurosurgery, Changzheng Hospital, Naval Medical University, Shanghai 200003, China; q@iiiz.org (X.Q.); asanghz@126.com (X.S.); 2Department of Radiology, PLA No. 905 Hospital, Naval Medical University, Shanghai 200050, China; 13918858728@139.com; 3Department of Medical Psychology, PLA No. 905 Hospital, Naval Medical University, Shanghai 200050, China; 4Department of Systems Medicine and Bioengineering, Houston Methodist Cancer Center, Houston, TX 77030, USA; stwong@houstonmethodist.org; 5Departments of Radiology, Pathology and Laboratory Medicine, Brain and Mind Research Institute, Weill Cornell Medicine, New York, NY 10065, USA

**Keywords:** cancer, methyladenosine, m^6^A, therapy, oncogenesis

## Abstract

**Simple Summary:**

Cancer remains a burden to the public health all over the world. An increasing number of studies have concentrated on the role of methyladenosine modifications on cancers. Methyladenosine modifications mainly include N6-methyladenosine (m^6^A), N1-methyladenosine (m^1^A), and 2’-O-methyladenosine (m^6^A_m_), of which dynamic changes could modulate the metabolism of RNAs in eukaryotic cells. Mounting evidence has confirmed the crucial role of methyladenosine modification in cancer, offering possibilities for cancer therapy. In this review, we discussed the regulatory role of methyladenosine modification on cancer, as well as their potential for treatment.

**Abstract:**

Methyladenosine modifications are the most abundant RNA modifications, including N6-methyladenosine (m^6^A), N1-methyladenosine (m^1^A), and 2’-O-methyladenosine (m^6^A_m_). As reversible epigenetic modifications, methyladenosine modifications in eukaryotic RNAs are not invariable. Drastic alterations of m^6^A are found in a variety of diseases, including cancers. Dynamic changes of m^6^A modification induced by abnormal methyltransferase, demethylases, and readers can regulate cancer progression via interfering with the splicing, localization, translation, and stability of mRNAs. Meanwhile, m^6^A, m^1^A, and m^6^A_m_ modifications also exert regulatory effects on noncoding RNAs in cancer progression. In this paper, we reviewed recent findings concerning the underlying biomechanism of methyladenosine modifications in oncogenesis and metastasis and discussed the therapeutic potential of methyladenosine modifications in cancer treatments.

## 1. Introduction

Cancer remains a considerable threat to public health worldwide. In 2022, the American Cancer Society estimates 1,918,030 new cancer cases and 609,360 cancer deaths [1]. An in-depth understanding of cancer biology is essential in identifying effective therapeutics strategies.

Cancer cells, similar to other eukaryotic cells, perform most of their functions through translated proteins by mRNAs. There are increasing numbers of chemical modifications observed on the bases of mRNA nucleosides. Methyladenosine modification refers to an additional methyl group inserted onto adenosine. mRNA maturation, exportation, degradation, and translation are largely regulated by methyladenosine modifications [2,3,4], including N6-methyladenosine (m^6^A), N1-methyladenosine (m^1^A), and 2’-O-methyladenosine (m^6^A_m_), among which the m^6^A modification is the most abundant [5,6]. In addition to mRNAs, methyladenosine modification also regulates noncoding RNAs (ncRNAs) [7] during oncogenesis, cancer metastasis, and therapeutic intervention [8]. The number of studies on RNA methylation in cancer has exploded since 2012. The role of methyladenosine modification has expanded from tumor genesis and metastasis to the regulation of tumor microenvironment, immunotherapy, drug resistance, etc. However, current reviews found in the literature do not cover the latest research findings. We present this review to fill this gap and to offer our perspective on the research direction in this field. We describe the structure of m^6^A, m^1^A, and m^6^A_m_ in eukaryotic cells; explicitly state how corresponding enzymes including methyltransferases (writers), demethylases (erasers), and readers modify RNAs in a dynamic manner, with focus on m^6^A-, m^1^A-, and m^6^A_m_ regulated RNAs in the oncogenesis and metastasis of cancer; and present our perspective on their therapeutic potential in cancer (Figure 1).

## 2. Methyladenosine Modifications in Eukaryotic RNAs

Nucleotides are the basic unit of nucleic acid. Since the 1950s, more and more chemical modifications of nucleotides on RNAs have been identified, enriching our knowledge about the structure, biogenesis, and function of RNAs to a great extent [9]. More than 60% of RNA chemical modifications are methylation modifications, which play a vital role in the RNA family [10,11]. In eukaryotic messenger RNAs (mRNAs), m^6^A is the most abundant type of chemical modification, affecting the structural stability, progress of translation, and regulation of the secondary structure of mRNAs [11,12]. While fifty years have passed since m^6^A was initially discovered in eukaryotic mRNA [13,14], it was not until the past decade that considerable progress has been made in understanding this modification.

In 2012, employing a combination of m^6^A-specific antibodies and gene sequencing techniques, transcriptome-wide analysis of m^6^A revealed the first glimpse of its landscape [10,15]. m^6^A is widely distributed in the mammalian transcriptome of more than 7000 mRNAs and 300 ncRNAs. However, the distribution of m^6^A modification is not random in most transcripts. The level of m^6^A modification at a certain site is determined by local sequence, followed by the secondary structure, and the distance from a “potential selected site” to the gene terminal area [16]. There is a fairly specific consensus motif, D(G, A, or U)R(G or A)ACH(A, C, or U) [9], at the m^6^A modification site, which is enriched in the 3′ untranslated region (3′UTR) as well as coding sequence (CDS) region, especially inside the long internal exons and around the area of stop codon [10,15] (Figure 2A). The 5’ end of introns was also found significantly enriched in m^6^A modification [12]. m^6^A could cause alterations to the mRNA structure, facilitating the binding of nuclear RNA-binding protein heterogeneous nuclear ribonucleoprotein C (HNRNPC) to intervene in the processing of precursor mRNA (pre-mRNA), which is a mechanism termed “m^6^A switch’’ [17]. Although the modification sites are evolutionarily conserved [10,15,16], there are some differences among different cells [12]. Besides sequence-specific sites, m^6^A modification seems more likely to be present in regions with secondary structures [12], although this remains controversial. m^6^A modifications are found in virtually all types of RNAs, including rRNAs, tRNAs [18], mRNAs, ncRNAs (microRNAs (miRNAs), circular RNAs (circRNAs), and long noncoding RNAs (lncRNAs)) [9], long intergenic noncoding (lincRNAs) [10], and small nucleolar RNAs (snoRNAs) [19] in various pathophysiological processes, such as hematopoiesis [12], germ cell genesis [20], and virus infection [21].

Another common base methylation modification is N1-methyladenosine (m^1^A). Dominissini et al., found that m^1^A modification is quite abundant in mammalian mRNAs around the starting codon, and its methyl donor is S-adenosylmethionine (SAM) [22]. m^1^A modification is evolutionarily conserved too, and positively correlates with translation efficiency and protein-level transcripts. m^1^A is mainly located in rRNAs and tRNAs in cytoplasm [6]. Unlike m^6^A, m^1^A can disturb the Watson–Crick base pairing [23], possessing the ability to inhibit the activity of translation by affecting ribosome scanning [6].

## 3. Writers, Erasers, and Readers in the Coregulation of the Dynamic Adjustment of Methyladenosine Modification

m^6^A is the most common type of reversible methyladenosine modification in mRNA [24] and has been a hot topic in epigenetic research for decades. Methylation or demethylation is realized with the help of methyltransferases (“Writers”) or demethylases (“Erasers”). Moreover, in the nucleus, these modifications could be bound by numerous nucleus-specific “Reader” proteins to affect the splicing and transport of mRNAs, and so on. When the transcript is transported to the cytoplasm, the m^6^A modification site will be bound to the cytoplasm-specific reader proteins, exerting its effect on a series of processes, such as mRNA metabolism and translation [25] (Figure 2B–D). The function of methyladenosine modification-related regulators was summarized in Table 1.

### 3.1. Writers

In mammalian cells, four distinct methyltransferases are known to be responsible for methylating m^6^A in different types of RNAs. Since most of the m^6^A sites in mRNAs are catalyzed by the methyltransferase-like 3 (METTL3)/METTL14 complex, this heterodimer is primarily described here [25].

The labeling of m^6^A in mRNAs depends mainly on a kind of methyltransferase complex (MTX) containing multiple components including the METTL3-METTL14 heterodimer (the core element), and several regulatory factors, for example, Wilm’s tumor 1-associated protein (WTAP) and KIAA1429, which are necessary for the integrity of enzyme function [28,29]. MTX is distributed in the nucleus rather than the cytoplasm, being selective to the sequences of the substrate, with no obvious preference for the structure of RNA [28]. In this heterodimer, METTL3 seems to be the only subunit comprising an active center with methyltransferase activity where there is a SAM binding site. METTL14 is not primarily responsible for catalyzing methyltranslation but provides an RNA-binding scaffold that increases enzyme activity by stabilizing the conformation of METTL3. In addition, this subunit seems to play a role in the recognition of bases, METTL3 cannot exercise catalytic activity alone in absence of METTL14. However, METTL14 may exhibit catalytic activity under certain circumstances, and the catalytic sites of METTL14 homologs vary among different species [26,27]. Although MTX has been well studied, there is a lack of research on its catalytic efficiency, which provides a direction for future research.

Labeling of m^1^A in mRNAs is known through MTX of TRMT6/TRMT61A complex and TRMT10C [30]. Phosphorylated C-terminal domain-interacting factor 1 (PCIF1), a cap-specific adenosine-N6-methyltransferase, catalyzes the formation of m^6^A_m_ on the first adenosine adjacent to m^7^G cap of mRNAs [30].

### 3.2. Erasers

A set of enzymes can demethylate RNA m^6^A, participating in the dynamic regulation of RNA activity. Fat-mass- and obesity-associated protein (FTO) was discovered as the first human m^6^A demethylase, existing in the nucleoplasm in a form of dot-like structure [24]. FTO is a homolog of AlkB family dioxygenases, catalyzes the reaction of oxidative demethylation of m^6^A in single-stranded RNAs (ssRNAs) instead of double-stranded RNAs (dsRNAs) in a Fe (II)- and alpha-ketoglutarate(α-KG)-dependent manner [31]. The activity of FTO is affected by pH in the environment with the optimum pH value being around 6.0. FTO also mediates the demethylation of m^6^A_m_ and shows the same enzyme activity for m^6^A_m_ and m^6^A on identical RNA sequences. It was reported that m^6^A is the most suitable substrate for FTO [31]. However, this is controversial because certain researchers consider FTO to mainly demethylate m^6^A_m_ on small nuclear RNAs (snRNAs), instead of demethylating m^6^A on mRNA, due to the cross-reaction of m^6^A antibodies with m^6^A_m_ [16,38]. Several antibody-independent testing techniques have been proposed for further research in this aspect [12,16,38]. In addition, m^1^A is a physiological substrate of FTO. FTO can demethylate m^1^A in tRNAs and RNAs with loop structure, lacking activity on linear ssRNAs and ssDNAs [31].

AlkB homolog 5 (ALKBH5), a member of the AlkB homolog (ALKBH) family, is the second identified RNA demethylase in mammals with a strong preference for ssRNAs, on which m^6^A modification is the main physiological substrate. ALKBH5 is diffusely dispersed in the nucleoplasm with an obvious granular aggregation, co-located well with some mRNA processing factors, such as phosphorylated serine/arginine-rich splicing factor 2 (SC35-PI), Smith antigen (SM), and alternative splicing factor/splicing factor 2(ASF/SF2) in nuclear speckles. Various cellular processes are under the control of ALKBH5, including assembly and modification of mRNA processing factors, translocation of transcripts, and the metabolism of RNAs [20]. Unlike FTO, ALKBH5 seems to be more specific in demethylating m^6^A rather than acting on m^6^A_m_ [32].

### 3.3. Readers

The first three proteins found to be m^6^A-binding proteins were two YT521-B homology (YTH) family proteins, namely YTH-domain-containing family protein 2 (YTHDF2), YTHDF3, and ELAVL1 (also known as HUR) [15]. Five kinds of YTH proteins are distributed in mammalian cells, divided into three classes: YTH-domain-containing protein (YTHDC)1, YTHDC2, and YTHDF protein family (YTHDF1-3). These five proteins contain the YTH domain that selectively binds to m^6^A [15,25,33].

Cytoplasm is the main repository of YTHDF1, YTHDF2, and YTHDF3. Earlier studies suggested that YTHDF1 could accelerate the translation process by promoting ribosome binding to target transcripts and increasing translation efficiency, whereas YTHDF2 plays a role in mRNA degradation by transporting mRNA with m^6^A modification to processing bodies (P bodies) and other RNA decay sites. YTHDF3 has both of the aforementioned effects; it promotes protein translation in coordination with YTHDF1, while also accelerating mRNA degradation together with YTHDF2 [33,34,35]. At the moment, there is a dispute over whether each YTHDF member performs totally distinct functions [25]. Given the high similarity of amino acid sequences among YTHDF proteins, it is not clear how they regulate diverse functions, and further research is necessary [12].

In contrast to its counterparts, YTHDC2 can be found in both the cytoplasm and nucleus. In addition to its DEAH-box domain and R3H core domain, the protein possesses two ankyrin repeat domains. YTHDC2 binds to 5′-3′ exonuclease by recognizing ankyrin repeat domains, which are independent of RNAs, indicating the possibility that it is involved in the regulation of RNA stability. YTHDC2 might also function in the translation process by liaising m^6^A-containing transcripts and ribosomes [36]. 

Similarly, YTHDC proteins could act as readers to combine with m^1^A sites, for which the affinity is weaker than that of m^6^A. m^1^A is bound directly by YTHDF-binding proteins 1–3 as well as the YTH domain of YTHDC1, but not YTHDC2. In addition, YTHDF1 seems to promote the translation of transcripts with an m^1^A modification while YTHDF2 accelerates the degradation of modified mRNAs [37].

## 4. Methyladenosine Modification: An Inseparable Part of Cancers

A growing number of studies have demonstrated the participation of methyladenosine modification in the formation (oncogenesis) and metastatic progression of cancer (Table 2). Overall, most m^6^A-related proteins show cancer-promoting properties. The exception is that METTL14 is predominantly cancer-inhibiting.

### 4.1. Writers in Oncogenesis

Research into oncogenesis, or tumorigenesis, has been employed for a long time to describe genetic changes contributing to the transformation of normal cells into cancer cells [98]. Incessant efforts have identified a series of oncogenes or tumor suppressor genes in leading to the genesis of cancers when aberrant [99]. By increasing the modification of m^6^A, METTL3 plays a crucial role in the oncogenesis of several cancers. Choe et al., indicated that METTL3 promotes the oncogenic transformation of human lung cancer cells by increasing the translation of oncogene-bromodomain-containing protein 4 (BRD4) in a eukaryotic initiation factor 3 (eIF3h)-dependent way [40]. METTL3 has been found to guarantee the stability of hexokinase 2 (HK2) and solute carrier family 2 member 1 (SLC2A1) via an m^6^A-insulin-like growth factor 2 mRNA-binding protein 2 (IGF2BP2)/3-dependent pathway, contributing to the tumorigenesis of colorectal cancer (CRC) [41]. Approximately 80% of colorectal cancer cases are likely to be affected by numerical and structural aneuploidy, especially broad copy number variations (BCNAs). The analyses conducted by Condorelli et al., revealed that the transcript levels of eIF3h (Chr8, q24.11) is 4–5-fold higher in Chr8q-, Chr13q-, and Chr20q-gained cancer samples, which might play a significant role in the recurrence of BCNAs in CRC [100,101]. Combining these together, it is reasonable to speculate that methyladenosine modification could impact CRC by modulating BCNAs in an eIF3h-dependent way. Further, METTL3 accelerates the oncogenesis of CRC by reducing the expression of yippee-like 5 (YPEL5) epigenetically in an m^6^A-YTHDF2-dependent way [42]. In bladder cancer, METTL3 reduces the expression of PTEN via expediting the maturity of pri-miR221/222, resulting in the malignant proliferation of cancer cells [43]. Moreover, the regulatory effect of METTL3 is prevalent in other types of cancers, including glioma [45,46], gastric cancer [47], hepatocellular carcinoma (HCC) [48], breast cancer (BRCA) [49], and acute myeloid leukemia (AML) [50].

METTL3 and METTL14 constitute a heterodimer MTX, in which METTL14 functions as the stabilizer for METTL3 to play the catalyzing reaction [102]. In previous studies, METTL14 has been regarded as a contributor to the oncogenesis of many cancers. For example, METTL14 plays an oncogenic role in AML by controlling leukemia stem cell self-renewal [51]. lnc942 upregulates the METTL14-mediated stability of mRNA including C-X-C motif chemokine receptor 4 (CXCR4) and Cytochrome P450 family 1 subfamily B member 1 (CYP1B1) in BRCA cells and thus promotes cell proliferation [52]. However, a growing number of studies have shown that METTL14 may have a predominantly negative regulatory role in oncogenesis. Yang et al., found that METTL14 suppresses the proliferation of CRC cells by inhibiting the transcription of oncogenic lncRNA X-inactive specific transcript (XIST) [53]. Similarly, the growth of CRC cells can be inhibited by METTL14 through changing the m^6^A modifications of miR-375/Yes-associated protein 1 (YAP1) pathway [54]. In gastric cancer, METTL14 was reported to be able to modulate the m^6^A modifications on circORC5 and thus abate the cell growth by the miR-30c-2-3p/AKT1 substrate 1 (AKT1S1) axis [55]. METTL14 was verified to restrain the skin oncogenesis induced by ultraviolet B (UVB) radiation via facilitating damage-specific DNA-binding protein 2 (DDB2)-mediated global genome repair in an m^6^A-dependent manner [56]. 

Recently, a study showed that WTAP upregulates the proliferation capability of HCC cells by suppressing the expression of ETS proto-oncogene 1 (ETS1) in a Hu-Antigen R (HuR)/p21/p27-dependent manner [57]. By mediating the stability of diaphanous-related formin 1 antisense RNA 1 (DIAPH1-AS1) in an m^6^A-dependent manner, WTAP promotes the formation of MTDH-LASP1 complex and the expression of LASP1, facilitating the cell growth of nasopharyngeal carcinoma [58].

Moreover, m^1^A methylation in a subset of tRNA is upregulated by TRMT6/TRMT61A, resulting in an increment of peroxisome-proliferator-activated receptor delta (PPARδ) translation, which in turn stimulates cholesterol synthesis and activates hedgehog signaling to promote the oncogenesis of HCC [59]. In bladder cancer, m^1^A upregulated by TRMT6/TRMT61A dysregulates the targetome of tRNA fragments, leading to the unfolded protein response and silencing of tumor suppressor genes [60].

### 4.2. Erasers in Oncogenesis

As an “eraser” of m^6^A, FTO removes the m^6^A modifications on RRACH motifs in mRNAs or lncRNAs in a Fe (II)/α-KG-dependent manner [103]. Oncogenesis is proven to be impacted by FTO. FTO is reported to demethylate m^6^A modifications in subtypes of AML, exerting their effect on the leukemogenesis [61,62]. In lung cancer, proliferation of cancer cells could be promoted by FTO-mediated upregulation of KRAS or MZF1 signaling [63,64]. FTO acts as a key factor in the tumorigenesis of oral squamous cell carcinoma by demethylating the translation initiation factor eIF4G1 [65]. Niu et al., reported that FTO lessens the pro-apoptosis effect of BCL2-interacting protein 3 (BNIP3) in an m^6^A-YTHDF2-dependent manner, contributing to the growth of breast cancer [66]. Furthermore, FTO was also deemed to promote oncogenesis by demethylating lincRNAs. For example, FTO upregulates LINC00022 epigenetically to promote esophageal squamous cell carcinoma [67]. However, FTO also exerts inhibitory effects on certain types of cancer [104]. FTO suppresses the oncogenesis of pancreatic cancer by inhibiting Wnt signaling as well as demethylating PJA2 [68]. In papillary thyroid cancer (PTC), Huang et al., found that FTO restrains the stability of Apolipoprotein E (APOE), repressing the glycolysis and growth of PTC in an m^6^A-IGF2BP2-mediated manner [69]. Another type of demethylase, ALKBH5, was found to promote oncogenesis of AML selectively in an m^6^A-dependent manner [70]. In addition, it is reported that ALKBH5 activates period circadian clock 1 (PER1), serving as a suppressor for pancreatic cancer in an m^6^A-YTHDF2 dependent way [71].

### 4.3. Readers in Oncogenesis

As mentioned above, m^6^A readers (YTHDF2, IGF2BP2, etc.) can work synergistically with m^6^A writers and erasers during the oncogenesis of various cancers. Some of them might call forth oncogenesis individually as well. A recent study showed that YTHDF1 increases eIF3C translation, which amplifies ovarian cancer oncogenesis [72]. In gastric cancer, YTHDF1 facilitates the expression of Wnt receptor frizzled 7 (FZD7), resulting in the upregulation of Wnt/β-catenin-related cell proliferation [73]. YTHDF2 is able to suppress the expression of AXIN1, which is an inhibitor of Wnt/β-catenin, resulting in the tumorigenesis of lung cancer [74]. Paternally expressed 10 (PEG10) is associated with proliferation, differentiation, and apoptosis of certain types of cancers, Zhang et al., reported that IGF2BP1 contributes to the progression of endometrial cancer by stabilizing paternally PEG10 mRNA in an m^6^A-dependent manner [75]. IGF2BP1 also promotes the expression of serum response factor (SRF, a transcriptional regulator) and stabilizes the expression of 35 kinds of SRF-mediated oncogenic genes including PDZ, LIM domain 7 (PDLIM7), and Forkhead box K1 (FOXK1), leading to the growth and invasion of ovarian, liver, and lung cancer [76].

### 4.4. Methyladenosine Modifications in Metastasis

An increasing number of studies have investigated the methyladenosine–tumor invasion–metastasis relationship. Yue et al., unveiled that METTL3 modulates the expression of zinc finger MYM-type containing 1 (ZMYM1) in an m^6^A-mediated manner, thus contributing to the metastasis of gastric cancer by recruiting the CtBP/LSD1/CoREST complex [77]. In prostate cancer, METTL3 improves the YTHDF2–HNRNPD-recognized degradation of ubiquitin-specific peptidase 4 (USP4) by mediating m^6^A modification on A2696 and then maintains the ubiquitin of ELAV-like RNA-binding protein 1 (ELAVL1), giving rise to the migration and invasion of cancer cells [78]. METTL3 enhances the maturation of pri-miR-1246 in an m^6^A-dependent manner, contributing to the metastasis of CRC via sprouty-related EVH1-domain-containing 2 (SPRED2)/MAPK signaling pathway [79]. In addition, the maturation of pri-miR-1246 mediated by METTL3 facilitates the metastasis of ovarian cancer via repressing the cyclin G2 (CCNG2) pathway [80]. Matrix metalloproteinase (MMPs) are proteolytic enzymes with the capability of disintegrating the membrane of extracellular band basement, potentiating the invasion of certain cancers [105]. According to Dahal et al., METTL3 reinforces the expression of MMP2, leading to an augmented invasion capacity of melanoma cells [81].

As for METTL14, most studies have identified its inhibitory effect on the metastasis of cancers. In CRC, METTL14 promotes the degradation of SRY-related high-mobility group box 4 (SOX4) in an m^6^A-dependent way, triggering the inhibition of metastasis of CRC via phosphatidylinositol 3-kinase (PI3K)/Akt signal [82]. During miRNA processing, METTL14 modulates the maturity of pri-miR-126 by interacting with DGCR8, repressing the metastasis of HCC [83]. Chen et al., reported that exon skipping of METTL14 induced by Cdc2-like kinases 1 (CLK1)/SR-like splicing factors5 (SRSF5) pathway tends to give rise to the metastasis of pancreatic cancer [84]. However, METTL14 was found to act as a fortifier for the metastasis of pancreatic cancer because it facilitates the turnover of p53 apoptosis effector related to PMP22 (PERP) mRNA [85]. Given the contradiction, the role of METTL14 in metastasis of cancers needs to be confirmed by further studies. Another methylase writer, WTAP, contributes to the migration and invasion of pancreatic cancer via stabilizing Fak mRNA and activating Fak-related pathways [86]. 

Erasers have been shown to have dual roles in the metastasis of cancers. In BRCA, FTO facilitates tumor cell migration and invasion by upregulating ADP ribosylation factor-like GTPase 5B (ARL5B) through demethylating the miR-181b-3p [87]. By decreasing the m^6^A modification level on integrin β1(ITGB1), FTO enables the metastasis of gastric cancer [88]. Conversely, FTO erases m^6^A modifications on metastasis-associated protein 1 (MTA1) mRNA and lowers the stability of MTA1 transcripts mediated by IGF2BP2, leading to the suppression of metastasis in CRC [89]. There have been numerous studies demonstrating that ALKBH5 inhibits cancer metastasis. For example, ALKBH5 serves as a suppressor for the metastasis of gastric cancer by downregulating the expression of protein kinase and membrane-associated tyrosine/threonine 1 (PKMYT1) in an IGF2BP3-m^6^A-mediated manner [90]. In addition, the suppressive effect of ALKBH5 was observed in prostate cancer, CRC, and non-small-cell lung cancers [91,92,93].

The regulatory role of methylated modification readers on cancer metastasis was also reported. Su et al., indicated that YTHDF1 binds with m^6^A modifications on EGFR, promoting EGFR expression to stimulate the metastasis of HCC after insufficient radiofrequency ablation [94]. YTHDC1 was found to be able to bind to m^6^A modifications on metastasis-associated lung adenocarcinoma transcript 1 (MALAT1), contributing to the expression of certain key oncogenes by maintaining the genomic binding sites of nuclear speckles, thus promoting metastasis [95]. According to Chang et al., YTHDF3 promotes breast cancer brain metastasis by inducing the upregulation of ST6 N-acetylgalactosaminide alpha-2,6-sialyltransferase 5 (ST6GALNAC5), gap junction protein (GJA1), and EGFR, whose transcripts are enriched in m^6^A modifications [96]. The inhibitory effect of YTHDF2 on the metastasis of lung adenocarcinoma was also investigated [97]. 

## 5. Theragnostic Potential of Methyladenosine Modifications in Cancer

The previous sections presented the non-negligible role of methyladenosine modifications in cancer oncogenesis and metastasis. Given the wide range of essential roles, many researchers have focused on their theragnostic potential on cancer, divided into three parts: acting as diagnostic markers and prognostic predictors, conditioning tumor microenvironment (TME) and immunotherapy, and modulating therapeutic resistance and self-renewal of cancer.

### 5.1. Acting as Diagnostic Markers and Prognostic Predictors

Early diagnostic and accurate prognostic biomarkers are both meaningful in helping clinicians make rational decisions in cancer treatment. The occurrence of cancer is often accompanied by alterations in the profiles of m^6^A expression. Alterations of m^6^A in tissues or in blood provide a possibility for early diagnosis of cancer. Huang et al., conducted liquid chromatography–electrospray ionization tandem mass spectrometry (LC-ESI-MS/MS) on nucleosides extracted from captured circulating tumor cells (CTC) and found an elevation of m^6^A modifications on mRNA that demonstrated the possibility of a noninvasive method for diagnosing cancer [106]. Significant upregulated m^6^A modification was detected in peripheral blood RNA in patients with breast cancer and was closely associated with the disease stages [107]. Receiver operating characteristic curve (ROC) analysis showed that the area under the curve (AUC) value of m^6^A is 0.887, larger than that of carbohydrate antigen 153 (CA153) as well as carcinoembryonic antigen (CEA) in the cohort. Ge et al., found that the m^6^A level has an AUC value of 0.929 in distinguishing gastric cancer from benign gastric disease, greater than that for CEA and carbohydrate antigen 199 (CA199) [108]. In lung cancer, the m^6^A level in leukocytes has an ascendant sensitivity and specificity in discriminating lung adenocarcinoma, lung squamous cell carcinoma, and non-small-cell lung cancer from healthy adults [109]. Through a support vector machine algorithm, Zhang et al., constructed a serum m^6^A-miRNA diagnostic signature to detect cancer ulteriorly [110]. They obtained 18 candidate miRNAs and tested them on 14,965 serum samples including 12 cancer types via training, internal validation, and external validation cohort. The results showed that this m^6^A-miRNA signature has a satisfactory diagnostic performance in identifying cancers, especially lung cancer, gastric cancer, and HCC [110]. However, current studies on the diagnostic potential of m^6^A tend to focus on overall expression profiles rather than on tumor-specific expression profiles. This creates a problem, even when changes in m^6^A modifications are detected in the blood, it is still challenging to localize specific tumors. Hence, it is imperative to find specific m^6^A-tagged markers for different cancers.

Prognosis prediction is important for cancer treatment as it can be used to guide treatment methods in order to achieve better therapeutic effects. Plenty of studies have focused on the prognostic value of methyladenosine modification. In gastric cancer, Wang et al., found that the expression level of METTL3, remarkably elevated in tumor tissue, is closely associated with poor prognosis [111]. Conversely, downregulation of METTL14 in rectal cancer is correlated with poor outcome in patients [112]. When it comes to lower-grade glioma, lung adenocarcinoma, breast cancer, and bladder cancer, risk models including m^6^A-mediated lncRNAs were built, revealing an effective value in predicting the prognoses [113,114,115,116]. More recently, Shen et al., conducted comprehensive analyses for the predictive value of 23 m^6^A regulators and 83 mRNAs and noncoding RNAs among 9804 pan-cancer samples [117]. The m^6^A signature they constructed exhibited a favorable performance in predicting the overall survival rate in 24 types of cancers. Zheng et al., found that m^1^A-regulating genes are correlated with advanced clinical stages and poor prognosis of pancreatic cancer, implying the possibility of m^1^A being an effective prognostic predictor [118]. Furthermore, Zhao et al., established an m^1^A signature model, by which the prognosis of HCC can be easily forecasted via evaluating the condition of tumor microenvironment (TME) [119]. 

These data implicate that methyladenosine modification could be implemented in the diagnosis and prediction of various cancers (Figure 3). Further basic and clinical research is needed to validate their effectiveness, which will pave the way for a better early-stage diagnosis as well as a more specific treatment for cancers. 

### 5.2. Conditioning Tumor Microenvironment and Immunotherapy

TME refers to the complex ecosystem comprising non-cancer host cells and non-cellular components, including endothelial cells, fibroblasts, immune cells, extracellular matrix, growth factors, cytokines, and so on [120]. As a breeding ground for neoplasms, TME potentiates the initiation, proliferation, invasion, and therapeutic resistance of cancer [121]. TME is involved in the complicated regulatory network of methyladenosine modifications, which, in turn, influences tumor development, progression, and efficacy of therapy.

Hypoxia is a characteristic of TME that could activate the progression of cancers through augmenting hypoxia-inducible factors (HIFs) [122]. So far, studies have confirmed that there is an interactive relationship between TME-related hypoxia and changes in m^6^A modification in tumors. For instance, breast cancer is associated with transcriptional activity of ALKBH5 mediated by HIF-1α and HIF-2α, leading to a decrease in m^6^A modification on NANOG mRNA and initiation of the tumor in the hypoxic TME [123]. In glioblastoma multiforme, Dong et al., found that demethylase ALKBH5 is associated with the recruitment of hypoxia-induced tumor-associated macrophages (TAMs); ALKBH5 deficiency suppresses the expression and secretion of CXCL8/IL8 [124]. Another study demonstrated that YTHDF1 facilitates the hypoxic adaption of non-small-cell lung cancer via Keap1-nuclear respiratory factor 2 (Nrf2)-Aldo-keto reductase family 1 member C1 (AKR1C1) axis [125]. In stomach cancer, m^6^A reader IGF2BP3 regulates the hypoxia-induced metastasis by increasing the expression of HIF-1α in an m^6^A-dependent manner.

The Warburg Effect refers to abnormally elevated glycolysis activation in cancer cells and the TME, a hallmark of tumor progression [126]. This metabolic dysfunction of cancer cells induces a hypoglycemic and acid condition of TME and further accelerates the progression of cancer [127]. METTL3 was found to upregulate glycolysis in CRC by stabilizing HK2 and SLC2A1 in an m^6^A-IGF2BP2/3-dependent manner, leading to cancer progression [41]. Conversely, FTO represses the glycolysis of PTC by destabilizing APOE, thus restraining the growth of cancer [69]. By modulating hypoxia and glycolysis in the TME, altering m^6^A modifications may serve as a therapeutic strategy in treating cancers, though more in-depth research is required. 

There is bidirectional crosstalk between the immune system and bone microenvironment, exerting a non-negligible effect on cancer [128]. Among them, the receptor activator of NF-kB (RANK)/RANK ligand (RANKL)/osteoprotegerin (OPG) pathway contributes to the interactions between cancer immunity and osteocytes [129,130]. For instance, the disturbance of RANKL/OPG ratio and osteoclastogenesis induced by cancer cells could facilitate the disruption of bone and implantation of metastases via downregulating the immune system pathway in a vicious circle [128]. If the bidirectional interaction between bone microenvironment and immune system can be effectively regulated during metastasis, it will be helpful for cancer treatment. More recently, Fang et al., reported that YTHDF2 alleviates osteoclast formation, bone resorption, and secretion of inflammatory cytokines in RANKL-primed osteoclast precursors by mediating NF-kB and MAPK signaling [131]. Further, m^6^A modification on circ_0008542 extracted from osteoblast exosomes could promote bone resorption by mediating the increasing RANK in osteoclast [132]. These findings implied the potential of m^6^A on the bidirectional crosstalk between the immune system and bone microenvironment. However, these current studies are limited to non-cancerous fields and more direct evidence is needed to demonstrate its effect on cancer.

TME has a characteristic of restraining immunoreaction and harboring immune-suppressive cells and cytokines, as well as immune checkpoint inhibitor receptors [133,134,135]. Further, TME accelerates the exhaustion of tumor killer cells, including CD8^+^ T cells and antigen-presenting cells (APCs) [136,137]. Numerous studies have indicated the regulatory role of m^6^A modification in the immunosuppression of TME. m^6^A regulates immune repression in the tumor microenvironment in two main ways: one is to regulate the infiltration of immune cells, and the other is to affect immune checkpoints inhibitors (ICIs). Xiong et al., found that METTL3 improves the m^6^A modification on Jak1 mRNA in tumor-infiltrating myeloid cells (TIMs) under the influence of lactate, promoting the immunosuppressive effect of TIMs [138]. Dong et al., reported that knockout of METTL14 in TAMs promotes the dysfunction of CD8^+^, imposing an immunosuppressive effect on the progression of CRC [139]. In intra-hepatic cholangiocarcinoma, ALKBH5 has been regarded to orchestra the expression of PD-L1 in TME in an m^6^A-mediated manner [140]. The modulating effect of m^6^A on immunosuppression of TME is expected to provide a new anchor for immunotherapy. Recently, METTL3 and 14 were found to weaken the potency of anti-PD-1 therapy by interfering with interferon-γ-signal transducer and activator of transcription 1 (STAT1)- interferon regulatory factor 1 (IRF-1) signaling [141]. This finding implies that knockdown of METTL3 and 14 would strengthen the immune response of anti-PD-1 immunotherapy. In breast cancer, Wan et al., found that METTL3/IGF2BP3 upregulates the expression of PD-L1 epigenetically, leading to a PD-L1-dependent T cell exhaustion and infiltration [142], suggesting METTL3/IGF2BP3-mediated m^6^A modification might be a novel immunotherapeutic target. In addition, depletion of FTO improves the sensitivity of melanoma to the anti-PD-1 blockade immunotherapy [143]. Deletion of ALKBH5 was also found to enhance the efficiency of anti-PD-1 immunotherapy via decreasing lactate and the infiltration of suppressive immune cells in TME [144]. Li et al., found that circNDUFB2 acts as an antitumor immune modulator via destabilizing IGF2BPs and activating RIG-I-MAVS signaling to facilitate the recruitment of immune cells in non-small-cell lung cancer (NSCLC) [145]. Furthermore, m^6^A scores were constructed in prostate cancer and pancreatic cancer to predict the prognosis and response to ICI therapy [146]. Meanwhile, Gao et al., provided an m^1^A score to evaluate the recruitment of immune cells in TME in CRC [147]. These findings together suggest that modulation of m^6^A/m^1^A could assist ICIs and forecast their curative effect, paving the way for an improvement of immunotherapy (Figure 4). Of note, m^6^A-associated proteins such as METTL3 tend to function globally instead of having a characteristic of tissue or RNA specificity. It is difficult to ensure that the altered methyladenosine modifications in specific RNA can be regulated without changing modifications in other RNAs. Therefore, how to specify the regulation of methyladenosine modifications is a complex problem for its application in cancer therapy.

### 5.3. Modulating Therapeutic Resistance and Self-Renewal of Cancers

Therapeutic resistance, such as radioresistance and chemoresistance, is one of the major challenges in cancer management [148]. A growing number of studies confirmed the role of m^6^A in therapeutic resistance (chemoresistance, drug resistance, and radioresistance), pointing to a new direction for future cancer therapy. Through targeting the myelocytomatosis oncogene (MYC)-miR-155/23a-Cluster-MXI1 pathway, the inhibition of demethylase FTO was able to potentiate the therapeutic effect of temozolomide in glioma [149]. In lung adenocarcinoma, METTL7B was found to mediate the resistance to EGFR-tyrosine kinase inhibitors (EGFR-TKIs) via methylating GPX4, HMOX1, and SOD1 mRNA. Jin et al., found that METTL3 induces the drug resistance of NSCLC via activating the MALAT1-miR-1914-3p-YAP axis in an m^6^A-dependent manner [150]. Meanwhile, Liu et al., unveiled that METTL3-mediated autophagy reduces the resistance of NSCLC cells to gefitinib via β-elemene [151]. In MCF-7 breast cancer xenograft model, METTL3 contributes to the adriamycin resistance by promoting the maturity of pri-microRNA-221-3p in an m^6^A-dependent manner [152]. WTAP promotes the chemoresistance to gemcitabine in pancreatic cancer by fortifying the stability of Fak mRNA [86]. Poly-(ADP-ribose) polymerase (PARP) inhibitor (PARPi) resistance remains an obstacle to the treatment of BRCA-mutated epithelial ovarian cancer (EOC). Fukumoto et al., found that m^6^A modification on FZD10 mRNA is correlated with the PARPi resistance via Wnt/β-catenin pathway [153]. Interestingly, PARP1 and Wnt/β-catenin pathways were found to be enriched in BRCA bearing the specific chromosomal aberrations (Chr1q gains and Chr16q losses) [154]. More studies are needed to investigate the possibility of methyladenosine modification modulating therapeutic resistance by regulating aberrations of chromosomes in cancers. ALKBH5-HOXA10 demethylates m^6^A modification on JAK2 mRNA, resulting in cisplatin resistance of EOC. Depletion of METTL3 enhances the sorafenib resistance of HCC through modifying the hypoxia TME via FOXO3-mediated autophagy [155]. In addition to the aforementioned drug resistance, m^6^A modification also exerts its effects on radioresistance. Wu et al., found that the stabilization of circCUX1 induced by m^6^A modification augments the radioresistance of hypopharyngeal squamous cell carcinoma via inhibiting the caspase-1 pathway [156]. These findings not only pinpoint the role of m^6^A modification on therapeutic resistance but also provide a new prospect for drug and radiation therapy for cancers.

In tumors, cancer stem cells are capable of sustaining pluripotency, promoting tumor progression, facilitating self-renewal, and proving resistance to drugs. Numerous studies have indicated that m^6^A modification acts as a modulator for the maintenance and self-renewal of cancer stem cells, providing a novel target for remedying tumors and eliminating the therapeutic resistance of cancer. Cui et al., demonstrated that the self-renewal and tumorigenesis of glioblastoma stem cells are largely dependent on m^6^A modification [157]. Dixit et al., also found that m^6^A reader YTHDF2 is a dependency of glioblastoma stem cells via stabilizing mRNAs of oncogene MYC and VEGFA [158]. In leukemia, inhibition of FTO attenuates the self-renewal of leukemia stem cells and suppresses immune checkpoint receptors, especially leukocyte immunoglobulin-like receptor B4 (LILRB4) [159]. ALKBH5 plays a crucial role in self-renewal and maintenance of leukemia stem cells and initiating cells in m^6^A-dependent ways [70]. In CRC cells, demethylation of cytoplasmic m^6^A_m_ by FTO enhances colorectal cancer stem-like phenotypes, potentiating the drug resistance of CRC [160]. What is more, FTO suppresses the self-renewal of cancer stem cells in ovarian cancer by inhibiting 3’, 5’-cyclic adenosine monophosphate (cAMP) signaling [161] (Figure 5). The above findings suggest that m^6^A can positively impact cancer therapy by regulating the self-renewal ability and drug resistance of tumor cells. For clinical applications, research on its specific effects needs to be strengthened.

## 6. Conclusions and Perspectives

Understanding methyladenosine modification in the development, regulation, and biological processes of diseases is a hot topic in recent years. Mounting evidence shows the important function of methyladenosine modification in controlling the fate of RNA. m^6^A modification is able to control the splice, exportation, stabilization, and translation of mRNA, ensuring normal physiological processes. Meanwhile, aberrant m^6^A modification on mRNAs leads to certain diseases, especially cancers. m^6^A writer, eraser, and reader proteins play a critical role in the oncogenesis and metastasis of glioma, breast cancer, HCC, gastric cancer, and CRC, etc. Thus, controlling the inordinate expression of m^6^A-related genes would be a novel strategy for the treatment of cancers. m^6^A modifications are also regarded as secondary structure modifications of ncRNAs. lncRNAs, circRNAs, and miRNAs themselves play vital roles in cancers [162]. Regulation of m^6^A modification to cancer can be achieved indirectly through their regulatory roles in ncRNAs. These characteristics render m^6^A an indispensable target in cancer research. This review systematically presents the role of methyladenosine modification in cancer and classifies its theragnostic effects into three categories, providing a new perspective for future research.

In view of the aforementioned properties, methyladenosine modification has the potential to be effective in cancer treatment. Considerable efforts have been made to implement the modulation of m^6^A in the remedy of cancer. On account of the maturation of methylated RNA immunoprecipitation sequencing (MeRIP-seq), the alteration profiles of a wide range of cancers have been screened. Among them, the upregulation of m^6^A modification in the peripheral blood RNA or leukocyte is able to diagnose certain types of cancer. In addition, the aberrantly modified m^6^A modification has the potency to predict the outcome of cancer. Regulating the m^6^A modification is supposed to control the TME of solid tumors, exerting an impact on the immunotherapy of tumors. Lastly, m^6^A modification adjusts the self-renewal of cancer stem cells and regulation of m^6^A is expected to have a unique value in the therapeutic resistance of tumors. 

Clinical application of targeting m^6^A in cancer requires further research. First, exploring suitable small-molecule inhibitors for the writers, erasers, and readers of m^6^A modification is desirable. Treatment might be improved through selective inhibition of m^6^A-related proteins in tumor tissues. More recently, Yankova et al., have identified a selective METTL3 catalytic inhibitor, STM2457, which could suppress the growth of AML [163]. Lee et al., reported that eltrombopag selectively inhibits the catalytic form of the METTL3-14 complex that can decrease the m^6^A modification on RNA in AML cells [164]. Meanwhile, two FTO inhibitors (FB23 and FB23-2) have been designed by Huang et al., exhibiting a selective inhibitory effect on the proliferation of AML [165]. Sabnis applied a novel small molecule ALKBH5 inhibitor with the activity of suppressing demethylase [166]. These advances pave the way for further development of cancer treatment. Second, as lack of specificity is a concern in epigenetic therapeutics, inhibition of methylases or demethylases may affect m^6^A modification in normal cells and interfere with their physiological functions. At the same time, the spectrum of m^6^A expression in different types of cancer may be extremely heterogeneous, and a unique m^6^A signature needs to be established for each type or subtype of cancer. Finally, an m^6^A-centered post-transcriptional editing needs to be investigated as an effective supplement to the existing gene editing therapy in cancer.

## Figures and Tables

**Figure 1 cancers-14-03195-f001:**
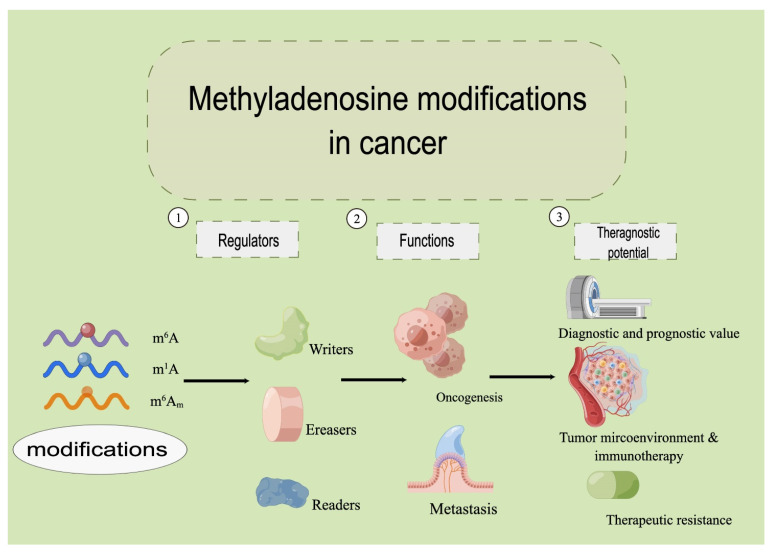
The overview of the methyladenosine modifications in cancer.

**Figure 2 cancers-14-03195-f002:**
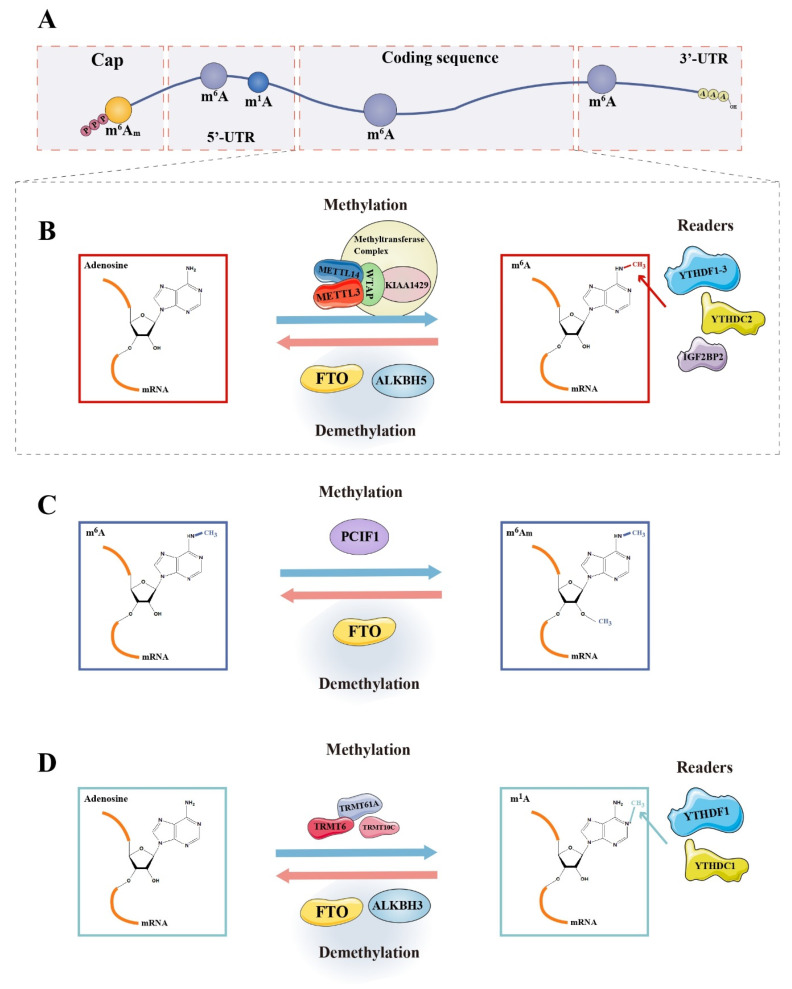
Dynamic changes in methyladenosine modifications on mRNA. (**A**) Distribution of m^6^A, m^1^A, and m^6^A_m_ modifications on mRNA; (**B**) methylases, demethylases, and readers of m^6^A modification; (**C**) methylases, demethylases, and readers of m^6^A_m_ modification; and (**D**) methylases, demethylases, and readers of m^1^A modification.

**Figure 3 cancers-14-03195-f003:**
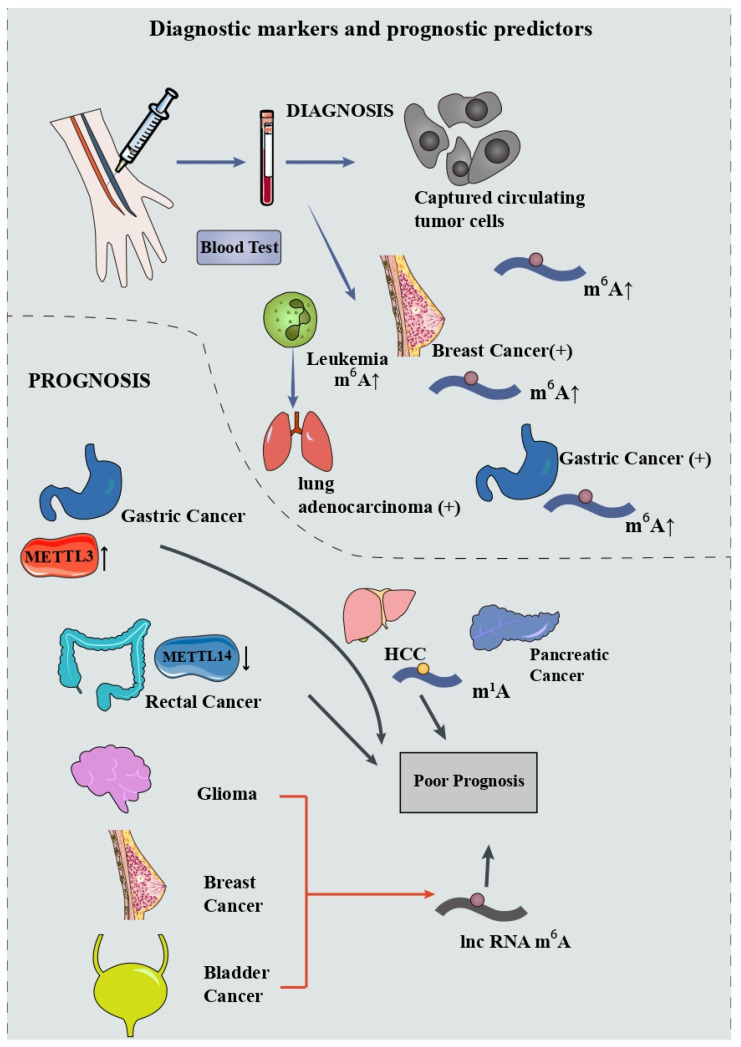
Methyladenosine modifications could act as diagnostic markers and prognostic predictors.

**Figure 4 cancers-14-03195-f004:**
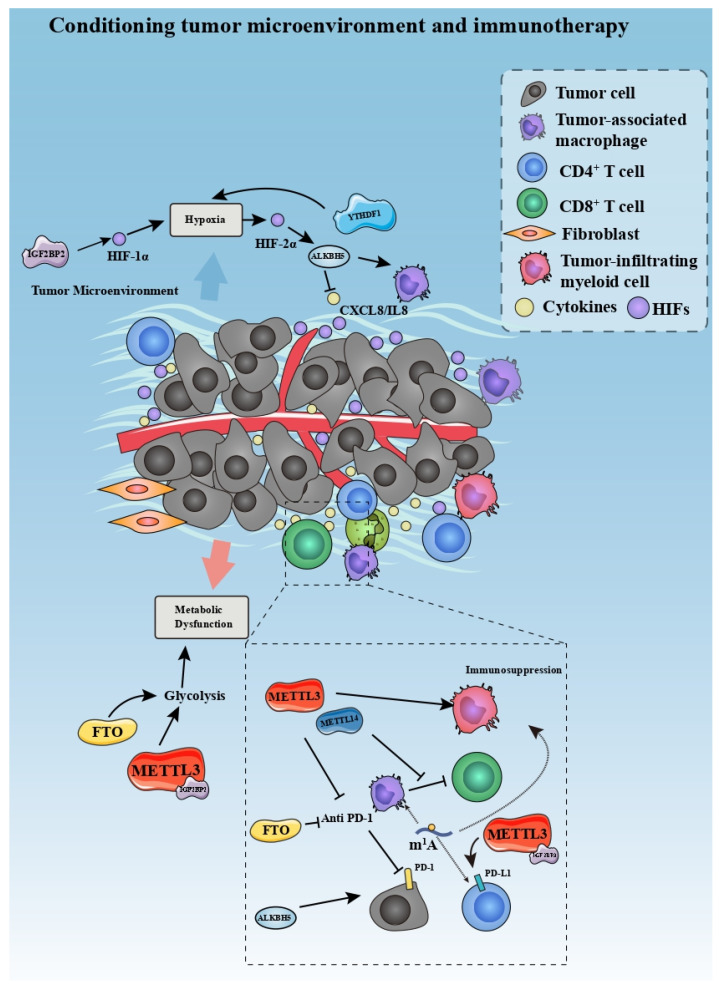
Methyladenosine modifications are able to condition tumor microenvironment and immunotherapy.

**Figure 5 cancers-14-03195-f005:**
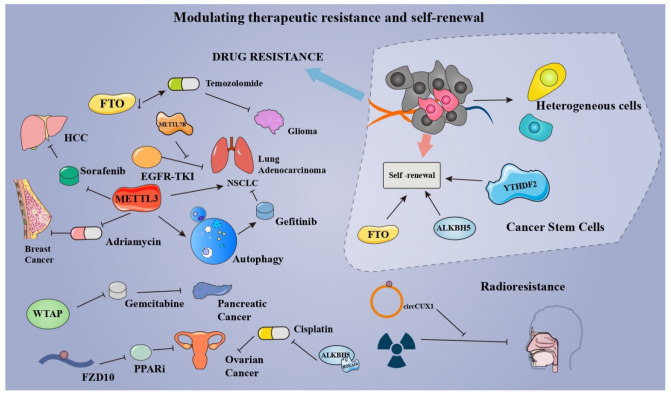
By modulating therapeutic resistance and self-renewal of cancers, methyladenosine modifications have the potential to mediate the efficacy of cancer therapy.

**Table 1 cancers-14-03195-t001:** The functions of m^6^A-related regulators.

Regulator	Target	Function	Reference
Writers			
METTL3	m^6^A	Binding to SAM; catalytic methylation	[26,27]
METTL14	m^6^A	Almost no methyltransferase activity; increases enzyme activity of METTL3 by stabilizing the conformation of METTL3; involved in recognition of bases	[26,27]
WTAP	m^6^A	Regulatory factors of MTX of METTL3-METTL14 complex	[28,29]
KIAA1429	m^6^A	Regulatory factors of MTX of METTL3-METTL14 complex	[29]
TRMT6/TRMT61A complex	m^1^A	Methyltransferases for mRNA in the cytosol	[6]
TRMT10C	m^1^A	Methyltransferases for mitochondrial mRNA	[6]
PCIF1	m^6^A_m_	Methyltransferases	[30]
Erasers			
FTO	m^6^A	Catalyzes the reaction of oxidative demethylation of m^6^A on ssRNAs	[31]
	m^6^A_m_	Demethylates m^6^A_m_ on snRNAs	[16]
	m^1^A	Demethylates m^1^A in RNAs with loop structure and tRNAs	[31]
ALKBH5	m^1^A	Demethylates specifically m^6^A on ssRNAs	[20,32]
Readers			
YTHDF1	m^6^A	Accelerate the translation process by promoting ribosome binding to target transcripts and increasing translation efficiency	[33]
YTHDF2	m^6^A	Affect mRNA degradation by transporting mRNA with m^6^A modification to P bodies and other RNA decay sites	[34]
YTHDF3	m^6^A	Promote protein translation in coordination with YTHDF1; accelerate mRNA degradation together with YTHDF2	[35]
YTHDC2	m^6^A	May involve in the regulation of RNA stability by binding to 5′-3′ exonuclease; affecting translation process by liaising m^6^A-containing transcripts and ribosomes	[36]
YTHDF1	m^1^A	Promote the translation of transcripts with an m^1^A modification	[37]
TYHDF2	m^1^A	Accelerate the deg-radiation of modified mRNAs	[37]

Writers, methyltransferases of methyladenosine modifications; erasers, demethylases of methyladenosine modifications; readers, recognizing & binding proteins of methyladenosine modifications.

**Table 2 cancers-14-03195-t002:** The emerging roles of methyladenosine modification in cancer.

Regulator	Modification	Role in Cancer	Cancer Type	Functional Pathway	References
METTL3	m^6^A	Oncogenesis	Human cancer cells	EGFR, TAZ	[39]
METTL3	m^6^A	Oncogenesis	Human lung cancer cells	BRD4, eIF3h-dependent	[40]
METTL3	m^6^A	Oncogenesis	Colorectal cancer (CRC)	HK2, SLC2A1, m^6^A-IGF2BP2/3-dependent	[41]
METTL3	m^6^A	Oncogenesis	CRC	YPEL5, m^6^A-YTHDF2-dependent	[42]
METTL3	m^6^A	Oncogenesis	Bladder cancer cells	PTEN, pri-miR221/222	[43]
METTL3	m^6^A	Oncogenesis	Hepatocellular carcinoma (HCC)	UBC9/SUMOylated METTL3/SNAIL axis	[44]
METTL3	m^6^A	Oncogenesis	Glioma		[45,46]
METTL3	m^6^A	Oncogenesis	Gastric cancer		[47]
METTL3	m^6^A	Oncogenesis	HCC		[48]
METTL3	m^6^A	Oncogenesis	Breast cancer (BRCA)		[49]
METTL3	m^6^A	Oncogenesis	Acute myeloid leukemia (AML)		[50]
METTL4	m^6^A	Oncogenesis	AML	Self-renewal of leukemia stem cells, initiation of AML	[51]
METTL4	m^6^A	Oncogenesis	BRCA	CXCR4, CYP1B1	[52]
METTL4	m^6^A	Anti-Oncogenesis	CRC	XIST	[53]
METTL4	m^6^A	Anti-Oncogenesis	CRC	miR375/YAP1 pathway	[54]
METTL4	m^6^A	Anti-Oncogenesis	Gastric cancer	miR-30c-2-3p/AKT1S1 axis	[55]
METTL4	m^6^A	Anti-Oncogenesis	Skin oncogenesis induced by UVB	DDB2	[56]
WTAP	m^6^A	Oncogenesis	HCC	ETS1, HuR/p21/p27-dpendent	[57]
WTAP	m^6^A	Oncogenesis	Nasopharyngeal carcinoma	DIAPH1-AS1, MTDH-LASP1 complex, LASP1	[58]
TRMT6/TRMT61A	m^1^A	Oncogenesis	HCC	PPARδ, Cholesterol synthesis, Hedgehog signaling	[59]
TRMT6/TRMT62A	m^1^A	Oncogenesis	bladder cancer	Targetome of tRNA fragments, Unfolded protein response, Genes silence	[60]
FTO	m^6^A	Oncogenesis	AML		[61,62]
FTO	m^6^A	Oncogenesis	Lung cancer	KRAS ang MZF1 signaling	[63,64]
FTO	m^6^A	Oncogenesis	Oral squamous cell carcinoma	eIF4G1	[65]
FTO	m^6^A	Oncogenesis	BRCA	BNIP3, m^6^A-YTHDF2-dependent	[66]
FTO	m^6^A	Oncogenesis	Esophageal squamous Cell carcinoma	LINC00022	[67]
FTO	m^6^A	Anti-Oncogenesis	Pancreatic cancer	Wnt signaling, PJA2	[68]
FTO	m^6^A	Anti-Oncogenesis	Papillary thyroid cancer (PTC)	APOE, m^6^A-IGF2BP2-dependent	[69]
ALKBH5	m^6^A	Oncogenesis	AML		[70]
ALKBH5	m^6^A	Anti-Oncogenesis	Pancreatic cancer	PER1, m^6^A-YTHDF2-dependent	[71]
YTHDF1		Oncogenesis	Ovarian cancer	eIF3C	[72]
YTHDF1		Oncogenesis	Gastric cancer	FZD7	[73]
YTHDF2		Oncogenesis	Lung cancer	AXIN1	[74]
IGF2BP1	m^6^A	Oncogenesis	Endometrial cancer	PEG10 mRNA	[75]
IGF2BP1		Oncogenesis	Lung, Ovarian and Liver cancer	SRF, FOXK1, PDZ, PDLIM7	[76]
METTL3	m^6^A	Metastasis	Gastric cancer	ZMYM1, CtBP/LSD1/CoREST complex	[77]
METTL3		Metastasis	Prostate cancer	A2696, USP4, ELAVL1	[78]
METTL3		Metastasis	CRC	pri-miR-1246, SPRED2/MAPK signaling pathway	[79]
METTL3		Metastasis	Ovarian cancer	pri-miR-1246, CCNG2 pathway	[80]
METTL3		Metastasis	Melanoma cells	MMPs	[81]
METTL14	m^6^A	Anti-Metastasis	CRC	SOX4, PI3K/Akt signaling	[82]
METTL14		Anti-Metastasis	HCC	pri-miR-126, DGCR8	[83]
METTL14		Anti-Metastasis	Pancreatic cancer	CLK1/SRSF5 pathway	[84]
METTL14		Metastasis	Pancreatic cancer	p53, PERP mRNA	[85]
WTAP		Metastasis	Pancreatic cancer	Fak mRNA, Fak-related pathways	[86]
FTO		Metastasis	BRCA	miR-181b-3p, ARL5B	[87]
FTO	m^6^A	Metastasis	Gastric cancer	ITGB1	[88]
FTO		Anti-Metastasis	CRC	MTA1, IGF2BP2	[89]
AKJBH5	m^6^A	Anti-Metastasis	Gastric cancer	PKMYT1, IGF2BP3-m^6^A-mediated	[90]
AKJBH5		Anti-Metastasis	Prostate cancer, CRC and non-small-cell lung cancer		[91,92,93]
YTHDF1	m^6^A	Metastasis	HCC (after insufficient radiofrequency ablation)	EGFR	[94]
YTHDC1	m^6^A	Metastasis	Esophageal cancer cells (ESCCs)	MALAT1	[95]
YTHDF3	m^6^A	Metastasis	BRCA	ST6GALNAC5, GJA1 and EGFR	[96]
YTHDF2		Anti-Metastasis	Lung adenocarcinoma		[97]

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
