# Peer review of "Methyladenosine Modification in RNAs: From Regulatory Roles to Therapeutic Implications in Cancer"

_cancers, 2022, doi:10.3390/cancers14133195_

Round 1

Reviewer 1 Report

Within the review ‚Methyladenosine modification in RNAs: from the regulatory roles to the therapeutic implications in cancer’ Qu et al. describe methyladenosine modifications, their generation, functions of writers, erasers and readers with a focus on cancers. The review also covers their theragnotsic potential.  

The review is very comprehensive and covers numerous findings related to cancer. At some points it is difficult to follow because it lists different findings without true connection. However, to gain an overview it is very helpful. 

Specific comments: Figures are very small. Some of them are not eligible anymore such as figure 2. It would be better to divide it in three part and magnify each of them. In addition, abbreviations are only given at the end. It would be good to have abbreviations within the text, otherwise one always needs to go forth and back. A table on the different functions of the methyltransferases would be helpful. In table 1 references should be included. Table 1 is cut on the right side, because it is too big for one page. 

Author Response

Thank you for the constructive comments concerning our manuscript entitled “Methyladenosine modification in RNAs: from the regulatory roles to the therapeutic implications in cancer” (#cancers-1740370). Those comments are helpful for revising and improving our paper. We hoped our revised manuscript will meet the publication criteria. Major revisions of the paper in response to the review comments are as follows:

Responses to reviewers’ comments

Reviewer #1

Comment 1: Within the review ‚Methyladenosine modification in RNAs: from the regulatory roles to the therapeutic implications in cancer’ Qu et al. describe methyladenosine modifications, their generation, functions of writers, erasers and readers with a focus on cancers. The review also covers their theragnotsic potential. 

The review is very comprehensive and covers numerous findings related to cancer. At some points it is difficult to follow because it lists different findings without true connection. However, to gain an overview it is very helpful.

Reply:  Thank you for your comments. We have included an overview before each section. We also added comments in each section to indicate issues of existing studies and provide our perspective on the research direction. In addition, a graphical abstract was presented to provide a more visual representation of the full text (see Figure 1).

Comment 2: Specific comments:

Figures are very small. Some of them are not eligible anymore such as figure 2. It would be better to divide it in three part and magnify each of them.

Reply:  We reorganized figure 2 into three separate images and enlarged the text inside for a better visual effect (see Figures 3, 4, and 5).

Comment 3: In addition, abbreviations are only given at the end. It would be good to have abbreviations within the text, otherwise one always needs to go forth and back.

Reply: We have inserted abbreviations into the test in accordance with your advice.

Comment 4: A table on the different functions of the methyltransferases would be helpful.

Reply: We added a table that summarizes different functions of writers, erasers, and readers of methyladenosine modifications (see Table 1).

Comment 5: In table 1 references should be included. Table 1 is cut on the right side, because it is too big for one page.

Reply: We addressed this issue to better present the table.

Finally, we would like to thank you for the constructive comments that help improve the quality and presentation of this review paper.

Your sincerely

Hong Zhao & Ding Ren

Reviewer 2 Report

Xiaolin Qu et al. uncovered epigenetic mechanisms to be relevant in therapeutic implications for cancer.

Points to be addressed:

1) The rationale of why the authors came up with this review.

2) What is the information that is not exactly available that motivated the authors to come up with this information. What are the current caveats and how do the authors highlight the current research in answering them? If not they need to address in future directions.

3) The authors pinpoint the hypoxic tumour microenvironment as a crucial actor for metastatic progression: this reviewer misses some insights regarding the bidirectional cross-talk between bone and immune system could become a potential target for anticancer drugs. 

4) In the frame of this thinking, the crucial and paradigmatic role for reactive oxygen species in RANKL-induced osteoclast differentiation is a good example of cancer progression in a hypoxic environment and should be expanded (refer to PMID: 32064051) 

5) The authors need to highlight what new information the review is providing to enhance the research in progress.

Few minor

  • a graphical abstract o a flow-chart describing the authors' methodology and workflow and findings might be beneficial
  • A native speaker revision can be beneficial

Author Response

Dear reviewer:

Thank you for the constructive comments concerning our manuscript entitled “Methyladenosine modification in RNAs: from the regulatory roles to the therapeutic implications in cancer” (#cancers-1740370). Those comments are helpful for revising and improving our paper. We hoped our revised manuscript will meet the publication criteria. Major revisions of the paper in response to the review comments are as follows:

Reviewer #2

Comment 1: The rationale of why the authors came up with this review.

Reply: We added the rationale of why we came up with this review. It was described in the Introduction section: “The number of studies on RNA methylation in cancer has exploded since 2012. The role of methyladenosine modification has expanded from tumor genesis and metastasis to the regulation of tumor microenvironment, immunotherapy, drug resistance, etc. However, current reviews found in the literature do not cover the latest research findings. We present this review to fill this gap and to offer our perspective on the research direction in this field.”

Comment 2: What is the information that is not exactly available that motivated the authors to come up with this information. What are the current caveats and how do the authors highlight the current research in answering them? If not they need to address in future directions.

Reply: We have added an overview to each section to better introduce the content. At the end of each part, we made comments on the corresponding content to provide a better direction for future research.

Comment 3: The authors pinpoint the hypoxic tumour microenvironment as a crucial actor for metastatic progression: this reviewer misses some insights regarding the bidirectional cross-talk between bone and immune system could become a potential target for anticancer drugs.

Reply: Thank you for your comments. We added the following contents in section “Conditioning tumor microenvironment and immunotherapy”:

“There is bidirectional crosstalk between the immune system and bone microenvi-ronment, exerting a non-negligible effect on cancer[1]. Among them, the Receptor Ac-tivator of NF-kB (RANK)/RANK Ligand (RANKL)/Osteoprotegerin (OPG) pathway contributes to the interactions between cancer immunity and osteocytes[2,3]. For instance, the disturbance of RANKL/OPG ratio and osteoclastogenesis induced by cancer cells could facilitate the disruption of bone and implantation of metastases via downregulating the immune system pathway in a vicious circle[1]. If the bidirectional interaction between bone microenvironment and immune system can be effectively regulated during me-tastasis, it will be helpful for cancer treatment. More recently, Fang et al. reported that YTHDF2 alleviates osteoclast formation, bone resorption, and secretion of inflammatory cytokines in RANKL-primed osteoclast precursors by mediating NF-kB and MAPK signaling[4]. Further, m6A modification on circ_0008542 extracted from osteoblast exo-somes could promote bone resorption by mediating the increasing RANK in osteoclast[5]. These findings implied the potential of m6A on the bidirectional crosstalk between the immune system and bone microenvironment. However, these current studies are limited to non-cancerous field and more direct evidence is needed to demonstrate its effect on cancer.”

Comment 4: In the frame of this thinking, the crucial and paradigmatic role for reactive oxygen species in RANKL-induced osteoclast differentiation is a good example of cancer progression in a hypoxic environment and should be expanded (refer to PMID: 32064051).

Reply: We added relevant content according to your suggestion (see reply to Comment 3 for details).

Comment 5: The authors need to highlight what new information the review is providing to enhance the research in progress.

Reply: We have highlighted the novel information the review is providing in the section of “Conclusions and perspectives” as follows: “This review systematically presents the role of methyladenosine modification in cancer and classifies its theragnostic effects into three categories, providing a new perspective for future research.”

Comment 6: A graphical abstract or a flow-chart describing the authors' methodology and workflow and findings might be beneficial

Reply:  Agree. We included a figure in the abstract (see figure 1).

Comment 7: A native speaker revision can be beneficial

Reply: We have revised our manuscript with the help of a native speaker.

Reference:

  1. Gnoni A, Brunetti O, Longo V, Calabrese A, Argentiero A-l, Calbi R, et al. Immune system and bone microenvironment: rationale for targeted cancer therapies. Oncotarget 2020, 11(4): 480-487.
  2. Morony S, Capparelli C, Sarosi I, Lacey DL, Dunstan CR, Kostenuik PJ. Osteoprotegerin inhibits osteolysis and decreases skeletal tumor burden in syngeneic and nude mouse models of experimental bone metastasis. Cancer Res 2001, 61(11): 4432-4436.
  3. Wong SK, Mohamad N-V, Giaze TR, Chin K-Y, Mohamed N, Ima-Nirwana S. Prostate Cancer and Bone Metastases: The Underlying Mechanisms. Int J Mol Sci 2019, 20(10).
  4. Fang C, He M, Li D, Xu Q. YTHDF2 mediates LPS-induced osteoclastogenesis and inflammatory response via the NF-κB and MAPK signaling pathways. Cell Signalling 2021, 85: 110060.
  5. Wang W, Qiao S-C, Wu X-B, Sun B, Yang J-G, Li X, et al. Circ_0008542 in osteoblast exosomes promotes osteoclast-induced bone resorption through m6A methylation. Cell Death Dis 2021, 12(7): 628.

Finally, we would like to thank you for the constructive comments that help improve the quality and presentation of this review paper.

Your sincerely

Hong Zhao & Ding Ren

Reviewer 3 Report

The article is of interest and is well written. It could be accepted in the current version, however, minor comments is shown below that might benefit readers to make things more clear.

Simple summary: It was expected in this part to define the impact of such modifications after line 20.

Abstract: What cause such RNA modifications in cancer?

1.Introduction

Please define methyladenosine in line 40. How it differs from methylation of C?

Also, one or more sentence to explain writers, erasers, and readers would be helpful in line 46.

2.M6A modifications in eukaryotic RNAs

It is unclear if exist in microRNAs or circRNAs as well or not (line 76)

While demonstrating its putative enrichment in 3’UTR ad CDS (line 67), what is the mechanism of causing insult to the gene function and oncogenesis? Does m6A influence gene expression level or secondary structure thus its function?

3.Writers Erasers Readers

What is the rate of modifictions?

4. This section (#4) is better to be illustrated by a figure to summarize the contents reviewed.

Figures: Figure 2 needs improvement. Suggest using Biorender. Fonts invisible. Colors need to be more calm and indicative.

Author Response

Dear reviewer:

Thank you for the constructive comments concerning our manuscript entitled “Methyladenosine modification in RNAs: from the regulatory roles to the therapeutic implications in cancer” (#cancers-1740370). Those comments are helpful for revising and improving our paper. We hoped our revised manuscript will meet the publication criteria. Major revisions of the paper in response to the review comments are as follows:

Reviewer #3

The article is of interest and is well written. It could be accepted in the current version, however, minor comments is shown below that might benefit readers to make things more clear.

Comment 1: Simple summary: It was expected in this part to define the impact of such modifications after line 20.

Reply: We have defined the impact of such modifications after line 20 as follows: “Methyladenosine modifications mainly include N6-methyladenosine (m6A), N1-methyladenosine (m1A), and 2'-O-methyladenosine (m6Am), of which dynamic changes could modulate the metabolism of RNAs in eukaryotic cells. Mounting evidence has confirmed the crucial role of methyladenosine modification in cancers, offering more possibilities for cancer therapy.”

Comment 2: Abstract: What cause such RNA modifications in cancer?

Reply: We have discussed the causes of methyladenosine modifications in cancer in the section of “Abstract” as follows: “Dynamic changes of m6A modification induced by abnormal methyltransferase, demethylases, and readers can regulate cancer progression via interfering with splicing, localization, transla-tion, and stability of mRNAs. Meanwhile, m6A, m1A, and m6Am modifications also exert regula-tory effects on non-coding RNAs in cancer progression.”

Comment 3: Please define methyladenosine in line 40. How it differs from methylation of C? Also, one or more sentence to explain writers, erasers, and readers would be helpful in line 46.

Reply: We have defined methyladenosine modifications and explain writers, erasers, and readers in the section of “Introduction” as follows: “There are increasing numbers of chemical modifications observed on the bases of mRNA nucleosides. Methyladenosine modification refers to an additional methyl group inserted onto adenosine.”

“We describe the structure of m6A, m1A, and m6Am in eukaryotic cells, explicitly state how corresponding enzymes including methyltransferases (writers), demethylases (erasers), and readers modify RNAs in a dynamic manner, with focus on m6A, m1A, and m6Am regulated RNAs in the oncogenesis and metastasis of cancer, and present our perspective on their therapeutic potential in cancer (Figure 1).”

Comment 4: M6A modifications in eukaryotic RNAs. It is unclear if exist in microRNAs or circRNAs as well or not (line 76). While demonstrating its putative enrichment in 3’UTR ad CDS (line 67), what is the mechanism of causing insult to the gene function and oncogenesis? Does m6A influence gene expression level or secondary structure thus its function?

Reply: We have revised the content in the section of “Methyladenosine modifications in eukaryotic RNAs” as follows: “m6A could cause the alterations of mRNA structure, facilitating the binding of nuclear RNA-binding protein heterogeneous nuclear ribonucleoprotein C (HNRNPC) to inter-vene in the processing of precursor mRNA (pre-mRNA), which is a mechanism termed as “m6A switch’’ [1].”

“Besides sequence-specific sites, m6A modification seems more likely to be present in regions with secondary structures[2], although this is still controversial. m6A modifications are found in virtually all types of RNAs, including rRNAs, tRNAs[3], mRNAs, ncRNAs (mainly include microRNAs [miRNAs], circular RNAs [circRNAs], and long non-coding RNAs [lncRNAs])[4], long intergenic non-coding(lincRNAs)[5], small nucleolar RNAs (snoRNAs)[6] in various pathophysiological processes, such as hematopoiesis[7], germ cell genesis[8], and virus infection[9].”

Comment 5: Writers Erasers Readers: What is the rate of modifictions?       

Reply: The question you raised is novel, and we have reviewed the relevant information but have not found studies concerning the catalytic efficiency of writers, erasers, and readers of methyladenosine modifications. However, this question is constructive and provides a direction for future research. We have added a sentence in the section of “Writers’, ’Erasers’ and ‘Readers’ to co-regulate the dynamic adjustment of methyl-adenosine modification” as follows: “Although MTX has been well studied, there is a lack of research on its catalytic efficiency, which provides a direction for future research.”

Comment 6: This section (#4) is better to be illustrated by a figure to summarize the contents reviewed. Figures: Figure 2 needs improvement. Suggest using Biorender. Fonts invisible. Colors need to be more calm and indicative.

Reply: Admittedly, an additional figure summarizing the contents of the section (#4) will be more illustrative. However, we have summarized this part through a table, and it seems a little bloated if we added another picture. Meanwhile, we have made extensive changes to Figure 2. Figure 2 was separated into three independent figures and text and arrows in it were modified to make the figures more indicative (Figures 3, 4, and 5). In addition, a figure was added to the abstract to summarize the content of the review (Figure 1).

Reference

  1. Liu N, Dai Q, Zheng G, He C, Parisien M, Pan T. N(6)-methyladenosine-dependent RNA structural switches regulate RNA-protein interactions. Nature 2015, 518(7540): 560-564.
  2. Hu L, Liu S, Peng Y, Ge R, Su R, Senevirathne C, et al. mA RNA modifications are measured at single-base resolution across the mammalian transcriptome. Nat Biotechnol 2022.
  3. Schmidt W, Arnold HH, Kersten H. Biosynthetic pathway of ribothymidine in B. subtilis and M. lysodeikticus involving different coenzymes for transfer RNA and ribosomal RNA. Nucleic Acids Res 1975, 2(7): 1043-1051.
  4. Huang H, Weng H, Chen J. mA Modification in Coding and Non-coding RNAs: Roles and Therapeutic Implications in Cancer. Cancer Cell 2020, 37(3): 270-288.
  5. Meyer KD, Saletore Y, Zumbo P, Elemento O, Mason CE, Jaffrey SR. Comprehensive analysis of mRNA methylation reveals enrichment in 3' UTRs and near stop codons. Cell 2012, 149(7): 1635-1646.
  6. Linder B, Grozhik AV, Olarerin-George AO, Meydan C, Mason CE, Jaffrey SR. Single-nucleotide-resolution mapping of m6A and m6Am throughout the transcriptome. Nat Methods 2015, 12(8): 767-772.
  7. Zheng G, Dahl JA, Niu Y, Fedorcsak P, Huang C-M, Li CJ, et al. ALKBH5 is a mammalian RNA demethylase that impacts RNA metabolism and mouse fertility. Mol Cell 2013, 49(1): 18-29.
  8. Kim G-W, Siddiqui A. N6-methyladenosine modification of HCV RNA genome regulates cap-independent IRES-mediated translation via YTHDC2 recognition. Proc Natl Acad Sci U S A 2021, 118(10).

Finally, we would like to thank you for the constructive comments that help improve the quality and presentation of this review paper.

Your sincerely

Hong Zhao & Ding Ren

Round 2

Reviewer 2 Report

The authors have clarified several of the questions I raised in my previous review. Most of the major problems have been addressed by this revision. The manuscript can be accepted for publication. 

Author Response

We thank you very much for the comments and suggestions on our manuscript.